# Research

evolution

reinforcement, reproductive isolation, assortative mating, hybridization, guenons

**Author for correspondence:**
Sandra Winters
e-mail: sandra.winters@nyu.edu

†Present address: School of Biological Sciences, Life Sciences Building, 24 Tyndall Avenue, Bristol, BS8 1TQ, United Kingdom.

# Simulated evolution of mating signal diversification in a primate radiation

Sandra Winters† and James P. Higham

Department of Anthropology, New York University, New York, NY, USA

(iD) SW, 0000-0002-9561-5950; JPH, 0000-0002-1133-2030

Divergence in allopatry and subsequent diversification of mating signals on secondary contact (reinforcement) is a major driver of phenotypic diversity. Observing this evolutionary process directly is often impossible, but simulated evolution can pinpoint key drivers of phenotypic variation. We developed evolutionary simulations in which mating signals, modelled as points in phenotype space, evolve across time under varying evolutionary scenarios. We model mate recognition signals in guenons, a primate radiation exhibiting colourful and diverse face patterns hypothesized to maintain reproductive isolation via mate choice. We simulate face pattern evolution across periods of allopatry and sympatry, identifying the role of key parameters in driving evolutionary endpoints. Results show that diversification in allopatry and assortative mate choice on secondary contact can induce rapid phenotypic diversification, resulting in distinctive (between species) and stereotyped (within species) face patterns, similar to extant guenons. Strong selection against hybrids is key to diversification, with even low levels of hybrid fitness often resulting in merged populations on secondary contact. Our results support a key role for reinforcement by assortative mating in the maintenance of species diversity and support the long-proposed prehistorical scenario for how such striking diversity was produced and maintained in perhaps the most colourful of all mammalian clades.

## 1. Introduction

Understanding the evolutionary mechanisms that produce and maintain animal diversity is a key topic in evolutionary biology. One commonly invoked mechanism for generating phenotypic diversity occurs when populations are geographically isolated (allopatry), which can lead to divergence because of drift and local adaptation [1]. If populations then experience secondary contact, selection against hybrids can lead to reinforcement and further diversification via the evolution of mating signals and preferences that evolve under diversifying selection to maintain reproductive isolation [1–4]. This process can lead to rapid phenotypic diversification and has been suggested as an important driver of species diversity in a wide variety of animal taxa (e.g. birds [5,6]; amphibians [7]; insects [8]).

Despite widespread interest in uncovering the evolutionary history and selective pressures that have generated extant phenotypes, such analyses are complicated by the inability to observe past evolution directly. This is particularly true for long-lived taxa with extended life histories, in which observing microevolutionary changes across even a few generations can take decades. Evolutionary trajectories and historical selection pressures can be inferred based on characteristics of extant species, phylogenetic comparisons, genetics, and the fossil record, but these methods each have their own limitations. Another powerful tool for studying evolutionary history is to simulate evolutionary change across time to identify the scenarios most likely to have generated the phenotypic patterns observed today. Evolutionary simulations have been extensively employed to model the processes of phenotypic diversification and speciation (e.g. [9–13]), and can produce insights into the importance of key parameters in generating

and maintaining patterns of phenotypic variation. For instance, phenotypic diversification is promoted by characteristics like assortative mating, ecological specialization, and low hybrid fitness [10,14,15]. Simulations that are grounded in a particular biological system can generate insight into both general biological processes as well as the most probable evolutionary drivers of particular phenotypes.

The guenons (tribe *Cercopithecini*) are a primate radiation consisting of 25–38 recognized species [16–19] that range throughout sub-Saharan Africa. Guenons diverged from papionin primates approximately 11.5 million years ago (Ma), with the diversification rate within the clade increasing around 2.8 Ma [20–22]. Speciation in guenons is thought to have occurred primarily in allopatry and has been linked to climate cycles that induced repeated contraction and expansion of African forests [23–27]. During dry periods, forest-dwelling guenons would have been restricted to isolated forest refugia; upon forest expansion, previously isolated populations would experience secondary contact. Many such cycles occurred throughout the evolutionary history of the group, and this repeated shift between allopatry and sympatry resulting from changing biogeography is thought to have been a major driver of guenon diversification.

Extant guenon species exhibit high degrees of sympatry, with sympatric species often forming mixed-species groups in which up to six guenon species travel and forage together [28,29]. Despite the widespread capacity for hybridization across guenons, hybrids are rare in most natural circumstances [30–34], suggesting the existence of pre-mating barriers to reproduction. Guenons exhibit strikingly diverse face patterns, which are hypothesized to function as mate recognition signals that maintain reproductive isolation between species through interspecific mate choice [30,31,35–38]. Recent research has used computer vision techniques to identify major axes of variation in guenon faces, rendering these complex signals easier to decompose and analyse [36,–38]. This work has shown that guenon faces exhibit character displacement, with facial distinctiveness associated with the degree of sympatry across the clade [36], suggesting diversifying selection between sympatric species. Within species, guenon faces are highly stereotyped, with minimal variation associated with age, sex or seasons [37,39], suggesting stabilizing selection or a lack of genetic variation within species. Guenons look longer at conspecific faces compared to heterospecific faces, which shows that they attend to these signals and could indicate increased mating interest [38]. This combination of evidence supports the hypothesis that guenon face patterns function as mate recognition signals that promote reproductive isolation between species, yet to date, to our knowledge, no study has directly assessed whether face patterns influence interspecific mate choice or reproductive isolation. Guenon mating is difficult to study in the wild and the slow life history of this group makes multi-generational analyses of fitness infeasible. Instead, we use evolutionary simulations to identify the scenarios and selective pressures most likely to have generated the diverse face patterns observed in guenons today. While modelled on the guenon clade, our results are generalizable to other species that have experienced shifts between allopatry and sympatry; this research therefore helps to shed light on the types of evolutionary processes involved in phenotypic diversification and reproductive isolation.

To provide new insights into the role of reinforcement in creating and maintaining species via mating signals generally,

and to evaluate the proposed evolutionary history of this clade specifically, we simulated a variety of scenarios in which faces evolved within a multi-dimensional phenotype space generated based on extant guenon features. 'Guenon' individuals with clade-average facial features initially evolved in isolated allopatric populations then experienced secondary contact in sympatry. We systematically varied time spent in sympatry, mating patterns, fitness of hybrids, population encounter rates, and number of co-evolving populations across simulations to determine the effect of these parameters on face pattern evolution. Using this simulation approach, we aimed to identify the conditions under which face patterns diversify to yield faces that are distinctive between populations and stereotyped within populations, and this diversity is maintained or accentuated in sympatry. To do this, we measured three key aspects of face pattern variation across simulations: (i) face pattern diversification; (ii) face pattern distinctiveness between populations; and (iii) face pattern variability within populations. We also investigated (iv) the evolution of female mating biases, to determine whether females became more discriminating as faces diverged. We predicted that face patterns would diversify (i.e. diverge between populations) in allopatry due to genetic drift, with greater diversification occurring in longer periods of allopatry, and that this diversity would be maintained in sympatry under mate choice when hybrids were of low fitness but not in other scenarios. In addition, we predicted that mate choice and low hybrid fitness would lead to the evolution of increasingly distinctive faces (i.e. character displacement) between newly sympatric populations, and that mate choice would reduce variation in face patterns within populations due to stabilizing selection on these distinctive phenotypes. Finally, we predicted that females would be more likely to engage in mate choice as populations diverged.

## 2. Methods

### (a) Generating guenon face space

We quantified face pattern diversity in extant guenon species using a multi-dimensional phenotype space generated using eigenface decomposition [36–38,40], a technique that uses principal component analysis to identify key axes of variation (eigenfaces) in aligned face images and has correlates in mammalian visual processing systems [41–43]. In this face space, the average guenon face is at the centre, and each dimension of the space characterizes an axis of facial variation; for instance, the first dimension broadly characterizes overall face colour from dark to light [36]. Within face space, a given face can be represented as a point based on its weight along each dimension of facial variation ($n = 15$ in this study). Faces can be reconstructed from face space weights as the sum of the average face and each eigenface image multiplied by the relevant weight. We generated guenon face space based on a previously collected database of images representing 21 extant guenon species. For more details, see the electronic supplementary material, Supplementary Methods.

### (b) Simulating face evolution

We simulated the evolution of guenon face patterns under a variety of scenarios. All simulations involved groups of male and female 'guenons', each of which had a facial phenotype defined as a vector of weights on the fifteen axes of face space. These face space weights evolved across simulated evolutionary time based on the characteristics of each simulated world.

Simulations were run in MATLAB [44] on the High Performance Computing Cluster at New York University.

Simulations included five key parameters: proportion of time in sympatry (50 or 90% of generations), population encounter rate (25, 50, or 75% likelihood of encountering a member of the same population in sympatry), hybrid fitness (0, 2, 5, 10, 50, or 90% likelihood of hybrids contributing to the next generation), number of co-evolving populations (2–6), and type of mate choice. We modelled three types of female mate choice: (i) no mate choice in which females mate passively; (ii) average mate choice for faces similar to the average face of the females' population; and (iii) positive assortative mate choice for faces similar to a females' own face. For more details on each of these parameters, see the electronic supplementary material, Supplementary Methods. A range of evidence suggests that initial divergence in guenons occurred in allopatry [23–27], and all simulations began with allopatric populations that then transitioned to sympatry. In allopatry, mating was only possible within populations and there was no mate choice. We did not model mate choice in allopatry in order to allow populations to accumulate differences in face patterns as a result of random drift. Drift is only one mechanism by which populations might diverge in allopatry, and local adaptation to different environments and ecological communities is also likely to play a role [45,46]. Here we use drift as a simple mechanism to generate variation that relies on minimal assumptions; drift may generate fewer differences in appearance across populations than other potential mechanisms, rendering our simulations conservative. We ran simulations implementing all unique combinations of these variables ($n = 540$), with 28 replications of each combination, yielding 15 120 total simulated worlds.

Each simulation began by initializing the relevant number of populations of 1000 'guenons' (generation 0) with facial phenotypes around the centre of face space. Each population was initialized and maintained at 50% male and 50% female. Males had a quality term used to generate male mating skew, initialized as a random value between 0 and 1 (but not bounded by this range); females had a bias term that indicated their likelihood of engaging in mate choice, initialized between 0 and 0.1 (bounded between 0 and 1). From these initial populations, we simulated 20 000 generations of evolution under each scenario described above. Each new generation was populated by generating 'offspring' from the current population via matings between males and females. Full details of simulated reproduction procedures are presented in the electronic supplementary material, Supplementary Methods. Briefly, we generated mating pairs by cycling through females (females have equal opportunities for reproduction) and pairing each with randomly drawn males, with each male's probability of selection proportional to his quality (to incorporate male reproductive skew) and the relevant population encounter rate. Under no mate choice, a single draw selected the father. Under average mate choice and positive assortative mate choice, multiple draws (representing 10% of the total male population) were made, and the female selected the male whose face was closest either to her population average face (average mate choice scenarios) or to her own face (positive assortative mate choice scenarios). In average mate choice and positive assortative mate choice scenarios, each females' mating bias term determined her likelihood of engaging in mate choice. This process was repeated until sufficient offspring had been produced for each population. Infants with parents from the same population were always retained in the next generation; hybrid infants (whose parents were from different populations) were retained in proportion with the hybrid fitness parameter. Offspring inherited their facial phenotype (i.e. a set of face space weights) from their parents, with each weight inherited randomly from one parent (i.e. face space weights were recombined across generations) and subject to

mutation. This mechanism of inheritance treats face patterns as complex phenotypes generated based on multiple heritable components. The genetic underpinnings of guenon face patterns are unknown, but they are probably polygenic traits involving many genetic loci. Here, offspring inherit some heritable markers from each parent; each marker (each face space weight associated with an eigenface) influences the whole face to some extent, and together the markers inherited from the two parents produce a face pattern which is influenced by both parents, but which is not a simple blend of parental faces. The resulting offspring became the parents of the next generation, and the simulations iterated.

## (c) Calculating evolved face pattern and female mating bias metrics

We evaluated evolved faces based on locations in face space. Each face was defined by a feature vector denoting the position in face space; in this space, distance between points (faces) indicates the degree of similarity. In this study, we are interested in the overall pattern of variation across faces, not in the particular faces that evolve, and so we measured the spread of faces across the space. Average faces were generated for each population by calculating the mean value for each feature across all individuals. We generated three metrics to quantify face pattern evolution: (i) diversification across populations; (ii) distinctiveness between populations; and (iii) variability within populations. We also measured (iv) female mating biases.

(i) Face pattern diversification was assessed using $k$-means clustering of evolved faces, with $k$ set to the number of co-evolving populations in the simulation. Each face was assigned to a population based on the partitioning of face space into clusters. In some instances, the 'correct' population identifier (the arbitrarily assigned population number) was different from that produced by the cluster analysis (e.g. population 1 assigned to cluster 2, population 2 assigned to cluster 1). To reconcile this and to identify the most parsimonious cluster identities, we used an iterative approach in which each population was originally assigned to its most common cluster, then conflicting designations were reconciled by assigning disputed clusters to their most common populations; the latter was repeated until all clusters had unique population designations. We then calculated the proportion of correctly clustered faces in each population; this value measures the extent to which the face patterns of the population are discriminable from those of its neighbours, with higher values indicating increasing diversification.

(ii) Face pattern distinctiveness between populations was calculated for each population as the mean Euclidean distance between their own average face and the average faces of all other populations in the scenario. Larger distances (i.e. the population is further from others in face space) indicate increasingly distinctive faces between the population and its neighbours.

(iii) Face pattern variability within populations was calculated for each population as the mean dyadic Euclidean distance between all population members. Smaller distances (i.e. individuals are closer in face space) indicate increasingly similar faces within the population.

(iv) Female mating biases were measured for each population as the average of all female mating bias terms.

## (d) Statistical analyses

We compared simulation outcomes across differing conditions using generalized linear mixed models in a Bayesian framework with the posterior distribution generated using Markov chain

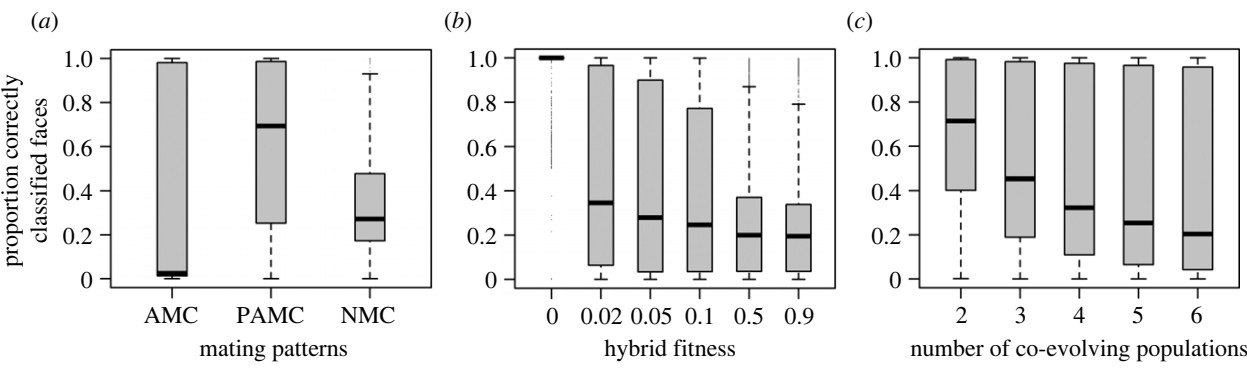

**Figure 1.** The proportion of correctly clustered faces across (*a*) mating patterns (AMC, average mate choice; PAMC, positive assortative mate choice; NMC, no mate choice), (*b*) hybrid fitness, and (*c*) the number of co-evolving populations. Populations are most diverse (reliably clustered) under positive assortative mate choice, low hybrid fitness, and a smaller number of co-evolving populations.

Monte Carlo (MCMC) simulations using the MCMCglmm package version 2.33 [47] in R v. 4.1.3 [48].

Simulation parameters were included in statistical models as fixed effects: type of mate choice (categorical), hybrid fitness (numerical: 0, 0.02, 0.05, 0.1, 0.5, and 0.9), proportion of evolution in sympatry (numerical: 0.5 and 0.9), population encounter rate (numerical: 0.25, 0.5, and 0.75) and number of co-evolving populations (numerical: 2–6); the simulated world (i.e. a unique identifier for each individual simulation) was included as a random effect in all statistical models to account for populations that evolved alongside one another. In this approach, repeated simulations under the same conditions ($n = 28$ for each combination of variables) serve as different observations. For all statistical models, we first determined whether the full model including all predictors was a better fit to the data than a null model including no fixed effects but the same random effects structure using Wald tests, implemented with aod version 1.3.2 [49] and coefplot2 version 0.1.3.2 [50] R packages. We then evaluated the significance of individual fixed effects based on their posterior mean values and associated pMCMC values. We tested the overall significance of mating pattern using Wald tests combining model coefficients; because each comparison across levels of mating pattern is made with respect to a baseline factor level, we re-levelled and re-ran models to generate comparisons across all mating patterns.

The three face pattern metrics and female mating bias were each included as the response variable in separate statistical models: (i) face pattern diversification, measured as the proportion of correctly clustered faces, was modelled using a binomial ('multinomial2' in MCMCglmm) error distribution, specified as the number of correctly and incorrectly clustered individuals; (ii) face pattern distinctiveness between populations, measured as the Euclidean distance between population average faces; (iii) face pattern variability within populations, measured as the mean Euclidean distance between faces within populations; and (iv) female mating biases, measured as the average female mating bias for each population, were all modelled using a Gaussian error distribution. In addition to the statistical model comparing female mating biases to the simulation parameters described above, we also compared female mating biases to face pattern distinctiveness between populations and face pattern variability within populations, with both metrics set as numerical predictors in separate statistical models. For more details on model construction and validation, see the electronic supplementary material, Supplementary Methods.

# 3. Results

## (a) Face pattern diversification
The model including all variables was a significantly better predictor of clustering accuracy of individuals by population

based on their evolved face patterns than a null model (Wald $\chi^2 = 4162.4$, $p < 0.001$). Clustering accuracy was highest under positive assortative mate choice, intermediate under no mate choice, and lowest under average mate choice (overall effect: Wald $\chi^2 = 739.3$, $p < 0.001$; no mate choice versus average mate choice: posterior mean $= -0.457$, pMCMC $< 0.001$; no mate choice versus assortative mate choice: posterior mean $= 1.663$, pMCMC $< 0.001$; average mate choice versus assortative mate choice: posterior mean $= 2.122$, pMCMC $< 0.001$; figure 1). Higher clustering accuracy was associated with lower hybrid fitness (posterior mean $= -5.607$, pMCMC $< 0.001$; figure 1) and fewer numbers of co-evolving populations (posterior mean $= -0.441$, pMCMC $< 0.001$; figure 1). Proportion of time evolving in sympatry (posterior mean $= -0.111$, pMCMC $= 0.503$) and population encounter frequency (posterior mean $= 0.065$, pMCMC $= 0.693$) did not significantly impact clustering accuracy.

## (b) Face pattern distinctiveness between populations
The model including all variables was a significantly better predictor of distances between populations in faces space than a null model (Wald $x^2 = 3544.5$, $p < 0.001$). All variables were significant predictors of population distances. Distances between populations were greatest under no mate choice, intermediate under positive assortative mate choice, and lowest under average mate choice (overall effect: Wald $\chi^2 = 570.0$, $p < 0.001$; no mate choice versus average mate choice: posterior mean $= -3.011$, pMCMC $< 0.001$; no mate choice versus assortative mate choice: posterior mean $= -0.495$, pMCMC $< 0.001$; average mate choice versus assortative mate choice: posterior mean $= 2.524$, pMCMC $< 0.001$; figures 2 and 3). Greater distances between populations were associated with a greater proportion of evolution in allopatry (posterior mean $= -3.104$, pMCMC $< 0.001$), lower hybrid fitness (posterior mean $= -0.085$, pMCMC $< 0.001$; figure 2), higher likelihoods of encountering members of the same population (posterior mean $= 0.013$, $p < 0.001$), and greater numbers of co-evolving populations (posterior mean $= 0.137$, pMCMC $< 0.001$).

## (c) Face pattern variability within populations
The model including all variables was a significantly better predictor of face pattern variability within populations (mean distance between individual faces) than a null model (Wald $\chi^2 = 99210.0$, $p < 0.001$). Facial variation was greatest under no mate choice, intermediate under assortative mate choice, and

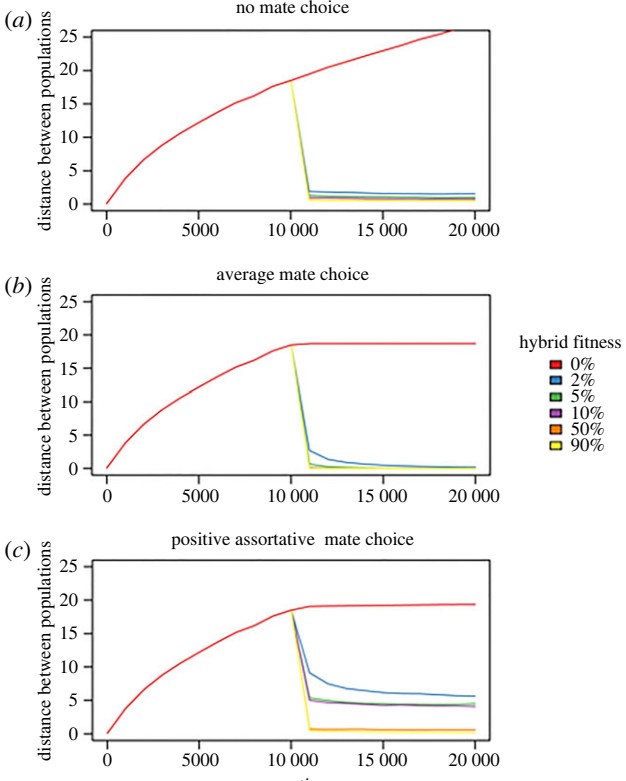

**Figure 2.** Mean Euclidean distances between populations across evolutionary time under different mating patterns and different degrees of hybrid fitness, with 10 000 generations in allopatry followed by 10 000 generations in sympatry. Under (*a*) no mate choice and (*b*) average mate choice, secondary contact leads to a hybrid swarm and facial distinctiveness between populations collapses unless hybrid fitness is zero. Under (*c*) positive assortative mate choice, some facial distinctiveness can be maintained under lower (2–10%) degrees of hybrid fitness. (Online version in colour.)

lowest under average mate choice (overall effect: Wald $\chi^2 = 97$ 305.1, $p < 0.001$; no mate choice versus average mate choice: posterior mean $= -7.350$, pMCMC $< 0.001$; no mate choice versus assortative mate choice: posterior mean $= -7.214$, pMCMC $< 0.001$; average mate choice versus assortative mate choice: posterior mean $= 0.136$, pMCMC $< 0.001$). Greater facial variation within populations was associated with a longer proportion of time evolving in allopatry (posterior mean $= -0.185$, pMCMC $< 0.001$), higher hybrid fitness (posterior mean $= 0.009$, pMCMC $< 0.001$), and greater numbers of co-evolving populations (posterior mean $= 0.280$, pMCMC $< 0.001$). Population encounter frequency was not a significant predictor of facial variation (posterior mean less than 0.0001, pMCMC $= 0.991$).

### (d) The evolution of female mating biases
When comparing the likelihood of females engaging in mate choice to all simulation parameters, the full model was not a significantly better predictor of female mating biases than a null model (Wald $\chi^2 = 4.500$, $p = 0.600$).

When comparing the likelihood of females engaging in mate choice to evolved face pattern distances, face pattern distinctiveness (mean Euclidean distance between each population average face and the average faces of all other populations) was a significant predictor of female mating biases (posterior mean $= 0.001$, pMCMC $< 0.001$), with higher female mating biases associated with greater distances

between populations; variation within populations (mean dyadic Euclidean distance between all population members) was not a significant predictor of female mating biases (posterior mean $= 0$, pMCMC $= 0.484$).

## 4. Discussion
Using evolutionary simulations, we show that diverse guenon face patterns can evolve under reinforcement when isolated populations experience secondary contact, a scenario that has long been proposed but not subjected to formal analysis. Across simulated evolutionary scenarios, low hybrid fitness and positive assortative mate choice tended to yield populations of 'guenons' with face patterns that were distinctive between species and stereotyped within species, similar to those of extant guenons. The proportion of evolution in sympatry, population encounter frequencies, and the number of co-evolving populations also influenced some aspects of face pattern variation. These results highlight the importance of low hybrid fitness and assortative mating in the maintenance of biological diversity under reinforcement and suggest that these variables may play a key role in driving phenotypic diversification in one of the most speciose and diverse primate radiations.

Our finding that low hybrid fitness promotes reproductive isolation is consistent with previous research [3,10], although reinforcement with gene flow is also possible [51,52]. Our results suggest that low hybrid fitness (likelihood of contributing to the next generation) is an important driver of face pattern diversity in guenons, as successful reproduction by hybrids often leads to a hybrid swarm and a lack of species maintenance on secondary contact. But critically, hybrid fitness does not need to be zero; diversification is still possible with low levels of gene flow when hybrids have low fitness. In our simulations, some populations achieved 100% correct clustering by population at hybrid fitness levels between 2 and 10%. Given that at least some guenons can produce fertile hybrids, low hybrid fitness is probably manifest via mechanisms such as poor survival (e.g. reduced immune capacity or antipredator behaviour), aberrant behaviours, lack of social integration, or unattractiveness to mates, rather than hybrid inviability or sterility. Our results are consistent with field observations of very low levels of hybridization between guenon species in most areas despite the capacity to do so; many of the observed instances of hybridization in guenons are the result of unnatural circumstances (e.g. captivity and degraded forests) [33]. An interesting case is that of red-tailed monkeys (*Cercopithecus ascanius*) and blue monkeys (*Cercopithecus mitis*), which are sympatric and commonly form mixed-species groups at many sites in East Africa [32]. In most forests, red-tailed × blue monkey hybrids are exceedingly rare or have not been observed despite substantial surveying; however, hybrids are common at Gombe National Park in Tanzania [32,33]. At Gombe, hybrids make up a substantial portion of the population, have been observed mating, and hybrid females have been observed nursing infants and juveniles. The guenons at Gombe exhibit a variety of phenotypes, including presumptive non-hybrid red-tailed monkeys and blue monkeys, $F_1$ hybrids, and backcross hybrids for each parental species. This blend of characteristics indicates that hybrids are successfully reproducing in this population and provides an example in which hybrids contributing to the next generation is associated with a breakdown of reproductive isolation and the resulting

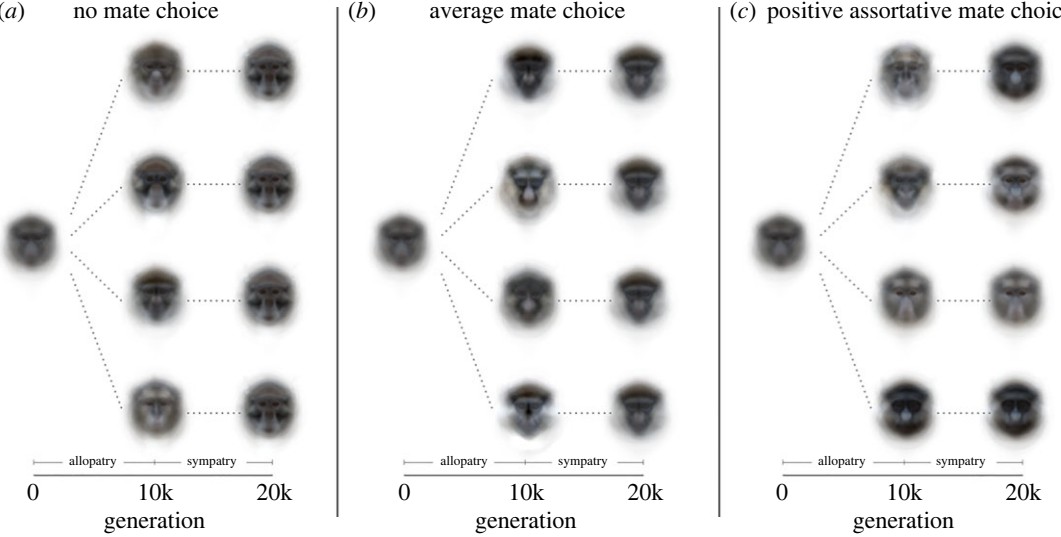

**Figure 3.** Examples of average faces of four populations evolving for 10 000 generations in allopatry followed by 10 000 generations in full sympatry (50% population encounter rate) with 2% hybrid fitness under three mating schemes. Distinctive face patterns are maintained in sympatry only for positive assortative mate choice. (*a*) No mate choice, (*b*) average mate choice, and (*c*) positive assortative mate choice. (Online version in colour.)

blending of phenotypes. It is unclear exactly what is driving this hybrid zone, but relatively low population densities and low conspecific mate availability have been suggested [32]. Overall, our results tying guenon diversification to low hybrid fitness are consistent with the low rates of hybridization in sympatric guenon species, and minimal postzygotic reproductive barriers in guenons [32], suggest the existence of strong pre-mating behavioural isolating mechanisms. Additional work documenting guenon mate choice and the behaviour of (usually rare) hybrids in the wild will be critical to understanding hybrid fitness in this group.

In our simulations, positive assortative mate choice generates diverse populations of 'guenons' with face patterns that are distinctive between populations and stereotyped within populations, similar to extant species [36,37,39]. Critically, assortative mating is associated with the most successful clustering of populations based on face patterns, the ultimate test of diversification. Yet assortative mating only generates intermediate levels of distinctiveness between populations and variation within populations. An important consideration is that our measure of distinctiveness does not account for population variance; under no mate choice (which yields higher distinctiveness, i.e. greater distances between populations in face space), there is large within-population variation, such that populations with more distinctive average faces may still overlap substantially within face space. Under positive assortative mating, within-population distances are relatively low and between-population distances are relatively high, suggesting truly distinctive faces. Retaining some facial variation within populations may also facilitate additional evolution of face patterns in sympatry under reinforcement. This may be why faces arising from average mate choice, which were less variable within populations, were also less distinctive between populations, and suggests that mate choice for species-average characteristics and the resulting stabilizing selection against novel phenotypes is unlikely to be involved in generating reproductive character displacement. Instead, this pattern is more likely under positive assortative mate choice in which there is increased scope for directional selection and phenotypic optimization. Our results highlight how mate choice can

generate the stereotyped phenotypes that are ideal for mating signals that function in reproductive isolation [39,53], but how it can also restrict additional diversification. By promoting mating between individuals exhibiting phenotypes at the same tail of the distribution, positive assortative mate choice can combine stabilizing and directional selection to optimize stereotyped signals. This underscores how the type of phenotypic preferences used as a basis for mate choice may be an important factor in promoting or restricting evolutionary diversification and phenotypic optimization. Future research should assess this possibility in more detail, including by investigating additional types of mate choice.

We also show that greater face pattern distinctiveness occurs when females are more likely to engage in mate choice, supporting a role for reinforcement in guenon diversification. These results are in line with more general conclusions that assortative mating plays a key role during reinforcement [1,3,15,54]. We did not observe any significant relationships between female mating biases and simulation parameters, which was unexpected. This suggests that stronger female mating biases tracked face pattern diversification, rather than the reverse, and perhaps an indirect relationship to simulation parameters was insufficient to generate links to these variables. Nonetheless, the positive relationship between female mating biases and face pattern distinctiveness between populations shows that the evolution of species-specific mating signals are viable and perhaps critical as reinforcement mechanisms in guenons. Studying guenon mating in natural settings is notoriously difficult, but will be key to disentangling interactions between signals and mate choice in this system.

Our simulations also show the impact of demographic variables on diversification. Across simulations, proportionally longer periods of evolution in sympatry (and therefore shorter periods of allopatry) resulted in more similar face patterns between and within populations, highlighting the importance of diversification in allopatry in generating novel phenotypes at early stages of divergence. Face pattern distinctiveness between populations increases with the likelihood of encountering same-population members, suggesting that access to conspecific mates may be important in driving

diversification and emphasizing the importance of considering varying degrees of sympatry. Future work could add a spatial component to these simulations to explore how the degree of population mixing and different geographical scenarios could play a role in evolutionary outcomes. The number of co-evolving populations also influences patterns of diversification; higher numbers of co-evolving populations are less discriminable (i.e. have a lower clustering accuracy across populations), but have greater distinctiveness between and variation within populations. The former is unsurprising, as more populations evolving within the same phenotype space could lead to phenotypic crowding. Yet guenon face space is very large, and the evolution of distinctive faces across species is clearly possible. It may be that longer periods of evolution are needed for larger numbers of sympatric populations to fully diversify. Greater distinctiveness between populations and variation within populations as more populations co-occur suggests a shift towards evolving unique interpopulation signals that are more easily distinguished, at the cost of stereotyped signals within populations that are often associated with mate recognition. Guenons are relatively unique in their propensity to form mixed-species groups with up to six con-generics, making them an ideal taxa in which to further investigate how the number of co-occurring lineages influences signal evolution.

Overall, this research provides important clues to the evolutionary drivers of phenotypic diversification in one of the most speciose and colourful primate radiations. We show that the proposed scenario in which guenon diversification was associated with repeated cycles of isolation and secondary contact in association with climactic cycling can lead to face pattern diversification when hybrids are of low fitness, but that hybrid fitness does not need to be zero. We also show that positive assortative mate choice based on

face patterns probably played a role in generating the diverse and stereotyped face patterns observed today. Higher female mating biases were associated with increased face pattern distinctiveness, suggesting a role for reinforcement in the generation and maintenance of diverse guenon face patterns. Our analyses model the evolution of face pattern diversity in guenons, but the overall conclusions are probably generalizable to any speciose taxa that has experienced shifts between allopatry and sympatry and is characterized by male reproductive skew. More broadly, our research demonstrates the types of processes that can generate or prevent the evolution and maintenance of phenotypic diversity in adaptive radiations with complex biogeographical histories.

**Data accessibility.** Code implementing simulations is available on GitHub: https://github.com/sandrawinters/guenon_evolutionary_simulations. Compiled simulation data are available as a Dryad Dataset [55] and code running statistical analyses is available on GitHub: https://github.com/sandrawinters/guenon_simulation_stats. Information is also provided in the electronic supplementary material [56].

**Authors' contributions.** S.W.: conceptualization, data curation, formal analysis, funding acquisition, software, visualization, writing—original draft, writing—review and editing; J.P.H.: conceptualization, funding acquisition, project administration, supervision, writing—review and editing.

Both authors gave final approval for publication and agreed to be held accountable for the work performed therein.

**Conflict of interest declaration.** We declare we have no competing interests.

**Funding.** This work was supported by the National Science Foundation (grant no. 1613378) and the American Society of Primatology. S.W. was supported by a National Science Foundation IGERT Fellowship and a New York University MacCracken Fellowship. New York University funded collection of the images used.

**Acknowledgements.** We thank Will Allen for access to guenon images. We thank two anonymous reviewers and one anonymous associate editor for comments which improved the manuscript substantively.

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
