## [Peer Review File · Proceedings of the Royal Society B: Biological Sciences]

Review History

RSPB-2021-0880.R0 (Original submission)

Review form: Reviewer 1

Recommendation

Major revision is needed (please make suggestions in comments)

Scientific importance: Is the manuscript an original and important contribution to its field?

Acceptable

General interest: Is the paper of sufficient general interest?

Acceptable

Quality of the paper: Is the overall quality of the paper suitable?

Acceptable

Is the length of the paper justified?

Yes

Should the paper be seen by a specialist statistical reviewer?

No

Do you have any concerns about statistical analyses in this paper? If so, please specify them explicitly in your report.

No

It is a condition of publication that authors make their supporting data, code and materials available - either as supplementary material or hosted in an external repository. Please rate, if applicable, the supporting data on the following criteria.

Is it accessible?

Yes

Is it clear?

Yes

Is it adequate?

Yes

Do you have any ethical concerns with this paper?

No

Comments to the Author

This is a review of "Simulated evolution of mating signal diversification in a primate radiation".

Here are my comments below:

1. The results do not model/simulate reinforcement per se since divergence only occurs when hybrids are 100% inviable (zero fitness). Thus, there is no gene flow between incipient populations. This is not reinforcement, it is RCD scenario ("kill the hybrids" scenario). Since that's the only time the simulations allowed for secondary contact to cause divergence, its very strange given that hybrids in mixed species flocks in guenons are apparently often viable and fertile (see Discussion)? Thus, the results do not explain what appears to occur in nature so this to me is the biggest weakness and discrepancy in this manuscript. Perhaps reinforcement is not the way to think about divergence and mating discrimination in guenons?
2. This leads me to ask: what does secondary contact really mean in this model in terms of migration rates? This was never specified; but if migration rate is 50% (full sympatry), then its way too high since most reinforcement scenarios assume that populations come back only partially and exchange migrants at some relatively low frequency. If its 50% migration rate than of course I wouldn't expect any divergence unless hybrids were 100% inviable. But with lower rates, one could expect reinforcement.
3. More realistic parameterization on hybrid fitness, mating system, and female mating preferences of guenons is needed. For instance, I was surprised that I didn't see absolute female mating preferences simulated with a given female having a certain facial preference since that's a typical approach in the literature. Migration rates between species need some further study and one needs to look into hybrids in detail and model their relationship/fitness compared to pure species.
4. In general, this was not a conceptually informative or novel study of RCD and it didn't really take advantage of the link between system and model. The main result of RCD with 0 hybrid fitness is not novel in speciation theory. Also they did not model divergence in mating preferences or sexual isolation per se and I think it would useful to do that in addition to facial diversification.

5. Finally, please explain better why no mate choice generates greater facial distances compared to average or assortative mate choice upon contact (line265 and Fig 5).

Review form: Reviewer 2

Recommendation

Major revision is needed (please make suggestions in comments)

Scientific importance: Is the manuscript an original and important contribution to its field?

Acceptable

General interest: Is the paper of sufficient general interest?

Acceptable

Quality of the paper: Is the overall quality of the paper suitable?

Marginal

Is the length of the paper justified?

Yes

Should the paper be seen by a specialist statistical reviewer?

No

Do you have any concerns about statistical analyses in this paper? If so, please specify them explicitly in your report.

Yes

It is a condition of publication that authors make their supporting data, code and materials available - either as supplementary material or hosted in an external repository. Please rate, if applicable, the supporting data on the following criteria.

Is it accessible?

Yes

Is it clear?

Yes

Is it adequate?

Yes

Do you have any ethical concerns with this paper?

No

Comments to the Author

In this manuscript the authors produce and simulate a model of mating signal (face pattern) evolution, informed by an empirical dataset on Guenon monkeys, to determine the most likely evolutionary scenario for the phenotypic variation seen today in the clade. More specifically, the simulations model scenarios of reinforcement after secondary contact, exploring several scenarios with and without mate choice and varying degrees of "hybrid fitness". The authors conclude that an scenario of zero or extremely low hybrid fitness and strong assortative mating is the most likely. While I found the manuscript interesting and potentially relevant for the journal, and the approach of informing the model with and applying it to a real dataset and

specific problem somewhat innovative, I see a few problems that I think should be addressed.

Major points

As a general comment, I think the exact insights produced by this model about either phenotypic diversification/speciation in general or for Guenons in particular should be expanded/clarified. More detail about the model is needed in the main text. Connections to previous work needs to be expanded.

One of the most important points to address is the lack of detail about how phenotypic evolution is modelled. More relevant details need to be in the main text (not in the supplement), but specially a clear justification of why phenotypic evolution was modelled this way is needed. I think the mechanism of “inheritance” (i.e. the mechanism of phenotypic evolution) is unrealistic and could be problematic. The infants inherit each full face feature (which the authors refer confusingly as a “genotype”) randomly either from their father or their mother, which is akin to a dominant Mendelian trait. That is, a complete set of trait values is identical to each parent. For a continuous, multidimensional, and highly complex trait as face pattern, I would say that a more realistic model - without having knowledge of the underlying (probably polygenic) genetic architecture - is one where the inherited face pattern is the mean of both parents plus a deviation (e.g. randomly drawn from a normal distribution) representing mutation. An extra concern is that as face features are derived from the eigen analysis of face patterns, variance is decreasing from one feature to the next, so the parent contributing (by chance) more of the first features will highly skew the overall phenotype of the infant towards its own face. This is clearly a “unorthodox” model of phenotypic evolution that needs some explanation and justification from the authors.

It's interesting that the authors decided to model a 15-dimensional “trait”, but this adds a lot of complexity to the model that I think is unexplored and its impact on model outcomes probably overlooked. Adding dimensions can lead to strange evolutionary dynamics in trait space (see for example Doebeli & Ispolatov *Am Nat* 2016). Also, dimensions are modelled as independent traits, but is it biologically realistic to assume that, for example, selection by assortative mating can operate effectively on 15 traits to lead to a significant divergence between two hybridizing populations? Perhaps this is the reason the authors see populations collapsing unless hybrid fitness is zero? I think that probably this deserves to be further explored.

And this takes me to the next important points. It is surprising that only for 0% fitness the process diversified in sympatry (Table 1). Could the authors provide more detail on this? I wonder how the construction of the model itself, and the choice of parameter values for the simulations could render it insensitive to diversification even when hybrid fitness is just 10% (but see above). Previous models and empirical work showed that diversification in sympatry with gene flow under assortative mating is possible. How are these results explained with respect to this previous evidence? (by the way I think references should be expanded).

Also, that diversification in sympatry with assortative mating is lower than with no mate choice is an unexpected result that needs explanation (Table 1). Also this might have to do with parameter choice (as above). The model seems insensitive to different levels of hybrid fitness. Perhaps assortative mating is not strong enough to counteract the high hybrid fitness. But this could be an artifact of the parameter space chosen. I think that further explanation and discussion of these results in this regard is needed.

I think that Fig 3 and 4 might reveal that the model could be badly parameterized or constructed. As soon as species enter into sympatry, and mate choice is allowed, populations immediately collapse to what seems to be zero variability. In the case of 0 hybrid fitness, different phenotypes are maintained (with no intrapopulation variability), but this parameter value not unexpectedly forces the model to do so. When fitness > 0, populations immediately collapse into a single phenotype, revealing that the parameter values are probably too high to reveal a response from

the model. It seems to me that a better equilibrium between “drift” by mutation, and selection by mate choice could be looked for through appropriate parameter values. Some discussion is granted, and I'm worried that the author's conclusion that zero hybrid fitness is needed to maintain diversity might be an artifact from model choices.

184 (...) we also assessed whether further diversification (character displacement) occurred after secondary contact in sympatry. We considered this to have occurred when mean distances between populations increased by at least one standard deviation during evolution in sympatry.

How and why was this threshold determined? I think that what happens after secondary contact should be compared to the outcome of the null process (i.e. allopatry), not to some arbitrary value. E.g. are distances at generation x larger for a process with sympatry than with pure allopatry?

251 Overall diversification was also observed in three scenarios where hybrid fitness was 10%, all occurring under assortative mate choice and with 50% of evolution occurring in allopatry.

This does not seem to be what Table 1 is showing? Please clarify

A relatively less important point is about male quality. Male quality is incorporated into the model but then how it impacts (or not) model outcomes is not mentioned or discussed. How does male quality itself evolve? There is strong selection for male quality so I'm not sure how male quality impacts the outcome of the model. Initial values are drawn from a uniform distribution? What's the average value after, say 10000x generations? I think male quality should be a relative value recalculated at each generation, otherwise the mean (absolute) value tends to evolve towards values of 1, which I don't think makes a lot of sense. In modeling the “devil is in the details” as people like to say, so in some cases adding “biological realism” adds little in terms of insights gained but a lot in complexity and interpretability of the model. I would like to see more detail and discussion on this.

Also, at the Supplement, line 108, it reads: “ (...) with each phenotypic parameter (face space feature weights and quality or bias terms) having a 0.5% chance of mutating.”

do you mean 0.5 instead of 0.5%? That would be a probability of mutating of 0.005 per generation, which seems to be very low.

Minor points

Line 78
or lack of genetic variation

- A very brief overview of eigenface decomposition would be useful for people not familiar with it.

134 From these initial populations, we simulated 20,000 generations of evolution using a genetic algorithm

What genetic algorithm? Please provide some detail

146 This models an average face model of face learning and discrimination, in which guenons cognitively encode different species' face patterns as the mean of all encountered examples.

Is there any justification for this?

174 in this space, distance between points (faces) indicates degree of similarity.

Euclidean distance? Squared Euclidean?

178 We considered populations within a scenario to have diversified when the mean distance between the faces of different populations was at least three standard deviations higher than the mean facial variation within populations.

Why was this cutoff chosen? Observation of behavior of the model? By eye? Please explain.

207 We compared evolved facial phenotypes across differing conditions using generalized linear mixed models (GLMMs) in a Bayesian framework

As I understand, what was compared was the distance between populations for the different scenarios (i.e. the degree of diversification). Please clarify.

269 The number of co-evolving populations was also a significant predictor of face distances (posterior mean = 0.0814, pMCMC = 0.026), with distances between faces decreasing with higher numbers of populations.

This might have to do with the fact that all populations start with the same phenotype. Adding populations will tend to reduce the distances as they will be more clumped in trait space.

296 (...) further diversification in sympatry and the evolution of stereotyped (within species) and distinctive (between species) faces were only observed under positive assortative mate choice.

This seems to be conflicting with results shown in Table 1 and described previously?

- The main text describing the model and simulations is almost a trimmed copy of the text in the supplement. Extra details provided there are hard to extract as the information is repeated.

- It would be interesting to see how interpopulation distances change through time for the different scenarios. A plot showing this should be simple to produce and is likely more informative than a table with arbitrary cutoff values for "diversification" or the figures with the faces or independent plots for the first three features. Faces are nice so I would keep them, but seeing the differences between the different outcomes is difficult, adding some sort of plot of distance through time would greatly help interpretation.

300 (...) these variables may play a key role in driving phenotypic diversification in one of the most speciose and diverse primate radiations.

Actually it seems that according to the model only zero fitness hybrids play a role in maintaining diversity after secondary contact.

Decision letter (RSPB-2021-0880.R0)

14-Jun-2021

Dear Dr Winters:

I am writing to inform you that your manuscript RSPB-2021-0880 entitled "Simulated evolution of mating signal diversification in a primate radiation" has, in its current form, been rejected for publication in Proceedings B.

This action has been taken on the advice of referees, who have recommended that substantial revisions are necessary. With this in mind we would be happy to consider a resubmission, provided the comments of the referees are fully addressed. However please note that this is not a provisional acceptance.

Sincerely,
Dr Locke Rowe
mailto:proceedingsb@royalsociety.org

Associate Editor
Comments to Author:

The manuscript has now been evaluated by two experts in the field. After careful evaluation of the reviews, I believe referees raised important concerns about the methodology and the support for the conclusions drawn.

Both reviewers agreed that this manuscript needs substantial work to address their concern and make a more impactful contribution. I believe the expert reviewers make valuable and constructive criticism that could significantly improve the manuscript.

Among other issues the reviewers brought-up the following concerns:

- Choice of parameter values for the models is not always consistent with aims (mechanisms of inheritance, hybrid viability etc)
- Lack of details about the way phenotypic evolution was modelled.
- Both reviewers were concerned that only a scenario where hybrids fitness was 0% (100% inviable hybrids) was considered for sympatry. In a related note, they raised concerns about the model being relatively insensitive to hybrid fitness variation, and the finding that diversification was lower with assortative mating than in the absence of mate choice, recommending the authors discuss this finding further and question whether it is the result of the parameter values chosen.

Reviewer(s)' Comments to Author:

Referee: 1

Comments to the Author(s)

This is a review of "Simulated evolution of mating signal diversification in a primate radiation".

Here are my comments below:

1. The results do not model/simulate reinforcement per se since divergence only occurs when hybrids are 100% inviable (zero fitness). Thus, there is no gene flow between incipient populations. This is not reinforcement, it is RCD scenario ("kill the hybrids" scenario). Since that's the only time the simulations allowed for secondary contact to cause divergence, it's very strange given that hybrids in mixed species flocks in guenons are apparently often viable and fertile (see Discussion)? Thus, the results do not explain what appears to occur in nature so this to me is the biggest weakness and discrepancy in this manuscript. Perhaps reinforcement is not the way to think about divergence and mating discrimination in guenons?

2. This leads me to ask: what does secondary contact really mean in this model in terms of migration rates? This was never specified; but if migration rate is 50% (full sympatry), then it's way too high since most reinforcement scenarios assume that populations come back only partially and exchange migrants at some relatively low frequency. If it's 50% migration rate then of course I wouldn't expect any divergence unless hybrids were 100% inviable. But with lower rates, one could expect reinforcement.

3. More realistic parameterization on hybrid fitness, mating system, and female mating preferences of guenons is needed. For instance, I was surprised that I didn't see absolute female mating preferences simulated with a given female having a certain facial preference since that's a typical approach in the literature. Migration rates between species need some further study and one needs to look into hybrids in detail and model their relationship/fitness compared to pure species.

4. In general, this was not a conceptually informative or novel study of RCD and it didn't really take advantage of the link between system and model. The main result of RCD with 0 hybrid fitness is not novel in speciation theory. Also they did not model divergence in mating preferences or sexual isolation per se and I think it would be useful to do that in addition to facial diversification.

5. Finally, please explain better why no mate choice generates greater facial distances compared to average or assortative mate choice upon contact (line 265 and Fig 5).

Referee: 2

Comments to the Author(s)

In this manuscript the authors produce and simulate a model of mating signal (face pattern) evolution, informed by an empirical dataset on Guenon monkeys, to determine the most likely evolutionary scenario for the phenotypic variation seen today in the clade. More specifically, the simulations model scenarios of reinforcement after secondary contact, exploring several scenarios with and without mate choice and varying degrees of "hybrid fitness". The authors conclude that an scenario of zero or extremely low hybrid fitness and strong assortative mating is the most likely. While I found the manuscript interesting and potentially relevant for the journal, and the approach of informing the model with and applying it to a real dataset and specific problem somewhat innovative, I see a few problems that I think should be addressed.

Major points

As a general comment, I think the exact insights produced by this model about either phenotypic diversification/speciation in general or for Guenons in particular should be expanded/clarified.

More detail about the model is needed in the main text. Connections to previous work needs to be expanded.

One of the most important points to address is the lack of detail about how phenotypic evolution is modelled. More relevant details need to be in the main text (not in the supplement), but specially a clear justification of why phenotypic evolution was modelled this way is needed. I think the mechanism of “inheritance” (i.e. the mechanism of phenotypic evolution) is unrealistic and could be problematic. The infants inherit each full face feature (which the authors refer confusingly as a “genotype”) randomly either from their father or their mother, which is akin to a dominant Mendelian trait. That is, a complete set of trait values is identical to each parent. For a continuous, multidimensional, and highly complex trait as face pattern, I would say that a more realistic model - without having knowledge of the underlying (probably polygenic) genetic architecture - is one where the inherited face pattern is the mean of both parents plus a deviation (e.g. randomly drawn from a normal distribution) representing mutation. An extra concern is that as face features are derived from the eigen analysis of face patterns, variance is decreasing from one feature to the next, so the parent contributing (by chance) more of the first features will highly skew the overall phenotype of the infant towards its own face. This is clearly a “unorthodox” model of phenotypic evolution that needs some explanation and justification from the authors.

It's interesting that the authors decided to model a 15-dimensional “trait”, but this adds a lot of complexity to the model that I think is unexplored and its impact on model outcomes probably overlooked. Adding dimensions can lead to strange evolutionary dynamics in trait space (see for example Doebeli & Ispolatov *Am Nat* 2016). Also, dimensions are modelled as independent traits, but is it biologically realistic to assume that, for example, selection by assortative mating can operate effectively on 15 traits to lead to a significant divergence between two hybridizing populations? Perhaps this is the reason the authors see populations collapsing unless hybrid fitness is zero? I think that probably this deserves to be further explored.

And this takes me to the next important points. It is surprising that only for 0% fitness the process diversified in sympatry (Table 1). Could the authors provide more detail on this? I wonder how the construction of the model itself, and the choice of parameter values for the simulations could render it insensitive to diversification even when hybrid fitness is just 10% (but see above). Previous models and empirical work showed that diversification in sympatry with gene flow under assortative mating is possible. How are these results explained with respect to this previous evidence? (by the way I think references should be expanded). Also, that diversification in sympatry with assortative mating is lower than with no mate choice is an unexpected result that needs explanation (Table 1). Also this might have to do with parameter choice (as above). The model seems insensitive to different levels of hybrid fitness. Perhaps assortative mating is not strong enough to counteract the high hybrid fitness. But this could be an artifact of the parameter space chosen. I think that further explanation and discussion of these results in this regard is needed.

I think that Fig 3 and 4 might reveal that the model could be badly parameterized or constructed. As soon as species enter into sympatry, and mate choice is allowed, populations immediately collapse to what seems to be zero variability. In the case of 0 hybrid fitness, different phenotypes are maintained (with no intrapopulation variability), but this parameter value not unexpectedly forces the model to do so. When fitness > 0 , populations immediately collapse into a single phenotype, revealing that the parameter values are probably too high to reveal a response from the model. It seems to me that a better equilibrium between “drift” by mutation, and selection by mate choice could be looked for through appropriate parameter values. Some discussion is granted, and I'm worried that the author's conclusion that zero hybrid fitness is needed to maintain diversity might be an artifact from model choices.

184 (...) we also assessed whether further diversification (character displacement) occurred after secondary contact in sympatry. We considered this to have occurred when mean distances between populations increased by at least one standard deviation during evolution in sympatry.

How and why was this threshold determined? I think that what happens after secondary contact should be compared to the outcome of the null process (i.e. allopatry), not to some arbitrary value. E.g. are distances at generation x larger for a process with sympatry than with pure allopatry?

251 Overall diversification was also observed in three scenarios where hybrid fitness was 10%, all occurring under assortative mate choice and with 50% of evolution occurring in allopatry.

This does not seem to be what Table 1 is showing? Please clarify

A relatively less important point is about male quality. Male quality is incorporated into the model but then how it impacts (or not) model outcomes is not mentioned or discussed. How does male quality itself evolve? There is strong selection for male quality so I'm not sure how male quality impacts the outcome of the model. Initial values are drawn from a uniform distribution? What's the average value after, say 10000x generations? I think male quality should be a relative value recalculated at each generation, otherwise the mean (absolute) value tends to evolve towards values of 1, which I don't think makes a lot of sense. In modeling the "devil is in the details" as people like to say, so in some cases adding "biological realism" adds little in terms of insights gained but a lot in complexity and interpretability of the model. I would like to see more detail and discussion on this.

Also, at the Supplement, line 108, it reads: "(...) with each phenotypic parameter (face space feature weights and quality or bias terms) having a 0.5% chance of mutating." do you mean 0.5 instead of 0.5%? That would be a probability of mutating of 0.005 per generation, which seems to be very low.

Minor points

Line 78
or lack of genetic variation

- A very brief overview of eigenface decomposition would be useful for people not familiar with it.

134 From these initial populations, we simulated 20,000 generations of evolution using a genetic algorithm

What genetic algorithm? Please provide some detail

146 This models an average face model of face learning and discrimination, in which guenons cognitively encode different species' face patterns as the mean of all encountered examples.

Is there any justification for this?

174 in this space, distance between points (faces) indicates degree of similarity.

Euclidean distance? Squared Euclidean?

178 We considered populations within a scenario to have diversified when the mean distance between the faces of different populations was at least three standard deviations higher than the mean facial variation within populations.

Why was this cutoff chosen? Observation of behavior of the model? By eye? Please explain.

207 We compared evolved facial phenotypes across differing conditions using generalized linear mixed models (GLMMs) in a Bayesian framework

As I understand, what was compared was the distance between populations for the different scenarios (i.e. the degree of diversification). Please clarify.

269 The number of co-evolving populations was also a significant predictor of face distances (posterior mean = 0.0814, pMCMC = 0.026), with distances between faces decreasing with higher numbers of populations.

This might have to do with the fact that all populations start with the same phenotype. Adding populations will tend to reduce the distances as they will be more clumped in trait space.

296 (...) further diversification in sympatry and the evolution of stereotyped (within species) and distinctive (between species) faces were only observed under positive assortative mate choice.

This seems to be conflicting with results shown in Table 1 and described previously?

- The main text describing the model and simulations is almost a trimmed copy of the text in the supplement. Extra details provided there are hard to extract as the information is repeated.

- It would be interesting to see how interpopulation distances change through time for the different scenarios. A plot showing this should be simple to produce and is likely more informative than a table with arbitrary cutoff values for "diversification" or the figures with the faces or independent plots for the first three features. Faces are nice so I would keep them, but seeing the differences between the different outcomes is difficult, adding some sort of plot of distance through time would greatly help interpretation.

300 (...) these variables may play a key role in driving phenotypic diversification in one of the most speciose and diverse primate radiations.

Actually it seems that according to the model only zero fitness hybrids play a role in maintaining diversity after secondary contact.

Author's Response to Decision Letter for (RSPB-2021-0880.R0)

See Appendix A.

RSPB-2022-0734.R0

Review form: Reviewer 1

Recommendation

Accept as is

Scientific importance: Is the manuscript an original and important contribution to its field?

Good

General interest: Is the paper of sufficient general interest?

Good

Quality of the paper: Is the overall quality of the paper suitable?

Good

Is the length of the paper justified?

Yes

Should the paper be seen by a specialist statistical reviewer?

No

Do you have any concerns about statistical analyses in this paper? If so, please specify them explicitly in your report.

No

It is a condition of publication that authors make their supporting data, code and materials available - either as supplementary material or hosted in an external repository. Please rate, if applicable, the supporting data on the following criteria.

Is it accessible?

Yes

Is it clear?

Yes

Is it adequate?

Yes

Do you have any ethical concerns with this paper?

No

Comments to the Author

I commend the authors for this meticulously revised version of their manuscript. It is a pleasure to review a properly structured revision, marked and with detailed responses. All my concerns have been properly and successfully addressed and now I consider the MS ready for publication.

Just caught two little things on the fly

Line 163 "based on a female rota" is this correct?

Line 170 missing punctuation

Author's Response to Decision Letter for (RSPB-2022-0734.R0)

See Appendix B.

Decision letter (RSPB-2022-0734.R1)

26-May-2022

Dear Dr Winters

I am pleased to inform you that your manuscript entitled "Simulated evolution of mating signal diversification in a primate radiation" has been accepted for publication in Proceedings B.

Data Accessibility section

Open Access

Your article has been estimated as being 9 pages long. Our Production Office will be able to confirm the exact length at proof stage.

Paper charges

Sincerely,

Dr Locke Rowe

Appendix A

Dear Editors,

Thank you for your consideration of our manuscript, “Simulated evolution of mating signal diversification in a primate radiation” (RSPB-2021-0880). We appreciate the thoughtful reviewer comments and welcome the opportunity to resubmit this manuscript.

Guenons are among the most brightly coloured and distinctive mammalian radiations; in this study, we ask the question: what evolutionary mechanisms might have generated the diverse face patterns observed in guenons today, and what insights might this process offer us into speciation processes more generally? To address these questions, we simulate multiple evolutionary scenarios to determine which is most likely to generate the pattern of variation we observe in extant guenons. In this resubmission, we have made multiple changes to our simulations and statistical analyses, which we feel have greatly improved this study. These include four key methodological changes: (1) we have added simulations of lower levels of hybrid fitness (2% and 5%, in addition to the original 0%, 10%, 50%, and 90%) to better resolve the impact on diversification at this end of the hybrid fitness spectrum, and have added a new parameter encoding different population encounter rates (as requested by reviewer 1); (2) we have replaced our binary classification of populations as ‘diversified’ or ‘not diversified’, which was based on an arbitrary threshold, with an approach that quantitatively measures population diversity using cluster analysis (as requested by reviewer 2); (3) we updated our statistical analysis of evolved distances in guenon face space to include hybrid fitness and to measure population-level variables (distances between populations and within populations, which quantify population distinctiveness and variability, respectively) rather than individual-level variables (e.g. dyadic distances between same- and different-population individuals), which more directly addresses our hypotheses and improves the interpretability of our results (as requested by reviewer 2); and (4) we have added an analysis of the evolution of female mating biases (as requested by reviewer 1). We have also made substantial changes to the manuscript text, based on these updated analyses and reviewer comments.

Our updated results suggest that mate choice and hybrid fitness are indeed key to driving evolutionary endpoints in populations of guenons which shift from allopatry to sympatry. Our simulations show that the evolution of diverse guenon face patterns that can be reliably clustered by population is most likely to occur under positive associative mate choice and low hybrid fitness, resulting in face patterns that are distinctive between species and stereotyped within species, as in existing guenon species. We now also show that females’ propensity to engage in mate choice is associated with increased distances between populations in face space, linking guenon face pattern diversity in sympatry to reinforcement. The overall conclusions of our study remain unchanged, but our updated analyses have provided a more nuanced picture of the likely evolutionary drivers of one of the most colourful and patterned mammalian radiations.

Below are our responses to reviewer comments, with our responses in bold. Line numbers refer to our updated (clean) manuscript; ESM refers to the Electronic Supplementary Materials.

We look forward to your response to this resubmission.

Best wishes,

Sandra Winters & James P. Higham

Associate Editor

Comments to Author:

The manuscript has now been evaluated by two experts in the field. After careful evaluation of the reviews, I believe referees raised important concerns about the methodology and the support for the conclusions drawn.

Both reviewers agreed that this manuscript needs substantial work to address their concern and make a more impactful contribution. I believe the expert reviewers make valuable and constructive criticism that could significantly improve the manuscript.

Among other issues the reviewers brought-up the following concerns:

- Choice of parameter values for the models is not always consistent with aims (mechanisms of inheritance, hybrid viability etc)

We have updated our analysis to include additional parameter values (new parameter encounter frequency, new parameter values for hybrid fitness) and have clarified our chosen mechanisms of inheritance. More details are in our responses to the reviewer comments below.

- Lack of details about the way phenotypic evolution was modelled.

We have added details to the text and justified our approach (lines 175-185; ESM lines 174-187). For a more detailed response, see below (Referee #2, Major Point #2).

- Both reviewers were concerned that only a scenario where hybrids fitness was 0% (100% inviable hybrids) was considered for sympatry. In a related note, they raised concerns about the model being relatively insensitive to hybrid fitness variation, and the finding that diversification was lower with assortative mating than in the absence of mate choice, recommending the authors discuss this finding further and question whether it is the result of the parameter values chosen.

This is an important point, and we thank both reviewers for their thoughtful and detailed comments. We now model six levels of hybrid fitness: 0, 2, 5, 10, 50, and 90%. We have also re-run our diversification analyses using cluster analysis, which is more objective, more directly answers the question “are these populations discriminable”, and provides greater resolution by directly quantifying the degree of discriminability (clustering accuracy), compared to our previous binary analysis (populations classified as diversified v. not). These new results show that while zero hybrid fitness leads to the greatest diversification (i.e., classification accuracy) across populations, lower levels of hybrid fitness can produce diversified populations as well, with hybrid fitness a statistically significant predictor of classification accuracy at the end of simulated evolution. Furthermore, our updated analyses of population distances show that

populations which evolve under assortative mating can be more reliably clustered by population than those that evolve under no mate choice or average mate choice. We still find that population averages are furthest apart under scenarios with no mate choice, which we address below (Referee #1, Point #5).

Reviewer(s)' Comments to Author:

Referee: 1

Comments to the Author(s)

This is a review of “Simulated evolution of mating signal diversification in a primate radiation”.

Here are my comments below:

1. The results do not model/simulate reinforcement per se since divergence only occurs when hybrids are 100% inviable (zero fitness). Thus, there is no gene flow between incipient populations. This is not reinforcement, it is RCD scenario (“kill the hybrids” scenario). Since that’s the only time the simulations allowed for secondary contact to cause divergence, its very strange given that hybrids in mixed species flocks in guenons are apparently often viable and fertile (see Discussion)? Thus, the results do not explain what appears to occur in nature so this to me is the biggest weakness and discrepancy in this manuscript. Perhaps reinforcement is not the way to think about divergence and mating discrimination in guenons?

We now model additional degrees of hybrid viability at the low end (newly added 2% and 5%), and we now show a more graded role of hybrid fitness rather than a sharp cut-off. Furthermore, our updated approach to characterizing diversification (cluster analysis rather than an arbitrary binary threshold) is more sensitive and generates more detailed results. All of our analyses now measure the full range of hybrid fitness values (rather than only analysing distances in face space for zero hybrid fitness). We also clarify that we are modelling hybrid fitness as the “likelihood of hybrids contributing to the next generation” (ESM lines 53-54), such that a hybrid that is viable and fertile but unable to secure a mate has zero fitness. We explain in the discussion how this is consistent with observations in extant guenons (lines 333-351). In our updated analyses we do see some divergence between populations when hybrid fitness > 0. The likelihood of evolving populations that can be reliably clustered by face pattern is highest under positive assortative mate choice, suggesting that reinforcement is indeed important to the evolution of guenon face pattern diversity. Furthermore, our new finding that higher female mating bias is associated with greater evolved distance between populations also suggests a role for reinforcement in guenon diversification. We have clarified this argument in the discussion (lines 381-392).

2. This leads me to ask: what does secondary contact really mean in this model in terms of migration rates? This was never specified; but if migration rate is 50% (full sympatry), then its way too high since most reinforcement scenarios assume that populations come back only

partially and exchange migrants at some relatively low frequency. If its 50% migration rate than of course I wouldn't expect any divergence unless hybrids were 100% inviable. But with lower rates, one could expect reinforcement.

Our goal here is not to model different geographic or migratory scenarios, and we originally assumed full sympatry (50% migration rate, as described above). However, we now model varying degrees of sympatry, with the probability of encountering members of the same population set to 100% (allopatry), 75% (same-population-biased sympatry), 50% (full sympatry), and 25% (different-population-biased sympatry) (described in lines 136-137; ESM lines 67-75). Our updated results show that encounter frequency does not influence overall population diversification (i.e., population clustering accuracy; lines 265-266), but that face pattern distinctiveness (distances between populations in face space) increase with the likelihood of encountering conspecifics (lines 279-280). This result is now discussed in the revised discussion section (lines 397-400). We agree that additional work focused particularly on degree of population mixing and different geographical scenarios would be useful; future work could add a spatial component to these simulations to explore this as well as other potential scenarios, such as parapatric speciation. We have now highlighted spatial/geographical modelling as a key area for future research (lines 400-402).

3. More realistic parameterization on hybrid fitness, mating system, and female mating preferences of guenons is needed. For instance, I was surprised that I didn't see absolute female mating preferences simulated with a given female having a certain facial preference since that's a typical approach in the literature. Migration rates between species need some further study and one needs to look into hybrids in detail and model their relationship/fitness compared to pure species.

We have added levels of hybrid fitness and population encounter rates (migration) to our simulations, which we agree has improved this study. We chose to model average mate choice (for the population-average face), and positive assortative mate choice, because both have straightforward potential mechanisms that are based on the existing phenotypes in the population. It is unclear how absolute female mating preferences would be implemented with respect to existing face patterns in these simulations, given that all populations start with the same phenotype. We agree that additional analyses of spatial dynamics and hybrid behaviour would be interesting and are important for understanding the evolution of diversity, however substantially expanding our simulations to model migration and hybrid relationships is beyond the scope of this study. We have now highlighted these as promising areas for future research (lines 400-402). We have nonetheless implemented many new scenarios for this revision in response to the comments of the reviewers.

4. In general, this was not a conceptually informative or novel study of RCD and it didn't really take advantage of the link between system and model. The main result of RCD with 0 hybrid fitness is not novel in speciation theory. Also they did not model divergence in mating preferences or sexual isolation per se and I think it would useful to do that in addition to facial diversification.

We have added an analysis of the evolution of females' propensity to engage in mate choice ('female mating bias'), and found that while this was not significantly related to

any of the parameters in our simulations, it is significantly associated with evolved face pattern distances between populations (lines 298-308). We have added a discussion of this result, highlighting that it supports a role for reinforcement in guenon evolution (lines 381-392). We have also added many additional simulated scenarios, including new levels of hybrid fitness and populations encounter rates. Our updated results show how face pattern diversification requires low, but not necessarily zero, hybrid fitness, with some simulated populations having very high clustering accuracy with 2-10% hybrid fitness.

5. Finally, please explain better why no mate choice generates greater facial distances compared to average or assortative mate choice upon contact (line 265 and Fig 5).

We agree that this result requires further clarification, and thank the reviewer for this comment. We ultimately find this result unsurprising because under no mate choice, face patterns continue to diversify (via drift) because there is no selection on facial phenotype. Under no mate choice, populations can remain distinctive in sympatry when hybrids are unfit, but otherwise populations merge. Mate choice, however, generates stabilizing selection on face patterns, which become stereotyped and tend to maintain the variation was generated via drift in allopatry. Stabilizing selection improves mate recognition, but also constrains further diversification of the phenotype (to a greater degree in average mate choice than positive assortative mate choice) such that the final evolved distances are smaller than under circumstances in which there is no mate choice. Furthermore (and perhaps most importantly), the distance between average population faces does not include measures of the variation within populations. Under no mate choice, there is large within-population variation, meaning that each population is a large cloud of points representing the faces of different individuals. Under these circumstances, the centre points of the species face-spaces might indeed be farther apart, but there might nonetheless still be substantial overlap between different populations. Under positive assortative mating, we observe within-population distances that are relatively low, and between-population distances that are relatively high, suggesting truly distinctive faces. This is highlighted by our (new) finding that positive assortative mate choice is associated with faces that are most reliably clustered by population (lines 257-262). We have clarified these arguments in the discussion (lines 356-365).

Referee: 2

Comments to the Author(s)

In this manuscript the authors produce and simulate a model of mating signal (face pattern) evolution, informed by an empirical dataset on Guenon monkeys, to determine the most likely evolutionary scenario for the phenotypic variation seen today in the clade. More specifically, the simulations model scenarios of reinforcement after secondary contact, exploring several scenarios with and without mate choice and varying degrees of “hybrid fitness”. The authors conclude that an scenario of zero or extremely low hybrid fitness and strong assortative mating is the most likely. While I found the manuscript interesting and potentially relevant for the journal, and the approach of informing the model with and applying it to a real dataset and specific problem somewhat innovative, I see a few problems

that I think should be addressed.

Major points

As a general comment, I think the exact insights produced by this model about either phenotypic diversification/speciation in general or for Guenons in particular should be expanded/clarified. More detail about the model is needed in the main text. Connections to previous work needs to be expanded.

We have re-written the main text describe model details more clearly (lines 128-185). We have also substantially expanded the scenarios we model in simulations; the addition of new levels of hybrid fitness and population encounter rates has produced important new results, and by simulating the evolution of female mating biases we are able to tie our results to mate choice and reinforcement more directly. We discuss how our results generate insights into the role of hybrid fitness (lines 322-253), mating patterns (lines 354-380), female mating biases (lines 381-392), and demographic variables (the proportion of time in sympatry, conspecific encounter rate, and the number of co-evolving populations; lines 393-415) in the evolution of diverse face patterns in guenons, and in signal diversification more generally.

One of the most important points to address is the lack of detail about how phenotypic evolution is modelled. More relevant details need to be in the main text (not in the supplement), but specially a clear justification of why phenotypic evolution was modelled this way is needed.

I think the mechanism of “inheritance” (i.e. the mechanism of phenotypic evolution) is unrealistic and could be problematic. The infants inherit each full face feature (which the authors refer confusingly as a “genotype”) randomly either from their father or their mother, which is akin to a dominant Mendelian trait. That is, a complete set of trait values is identical to each parent. For a continuous, multidimensional, and highly complex trait as face pattern, I would say that a more realistic model - without having knowledge of the underlying (probably poligenic) genetic architecture - is one where the inherited face pattern is the mean of both parents plus a deviation (e.g. randomly drawn from a normal distribution) representing mutation. An extra concern is that as face features are derived from the eigen analysis of face patterns, variance is decreasing from one feature to the next, so the parent contributing (by chance) more of the first features will highly skew the overall phenotype of the infant towards its own face. This is clearly a “unorthodox” model of phenotypic evolution that needs some explanation and justification from the authors.

Many thanks for this comment. We have re-structured the Methods and Supplementary Methods sections to provide more details regarding the mechanism of phenotypic evolution in the core text of the manuscript, and feel this updated structure has substantially improved the clarity of this our manuscript. We appreciate these concerns regarding the mechanisms of inheritance, however we feel that our approach is appropriate. Offspring inherit face space weights from their parents, which are numerical values that denote the location in face space along each axis of facial variation. These weights combine to yield a complex facial phenotype; e.g., in our approach, an offspring would not inherit it’s nose from one parent or another, but the weights of both parents would combine to yield a face that has a nose, the appearance of

which might be influenced by all 15 axes of variation (some of which came from the mother, some from the father). This approach is therefore more akin to inheritance of a mix of parental phenotypes than to discrete Mendelian inheritance of a whole face pattern from one parent or the other. This is “consistent not only with our expectations over how complex traits are inherited, but also with observations from real guenon hybrid species, which always display a mix of the facial features of both parental species” (lines 183-185). Furthermore, we argue that “encoding a single phenotype (the face) using a 15-dimensional space is akin to modelling a polygenic trait with multiple (unknown) genetic loci that influence the phenotype” (lines 180-182); face space weights (continuous variables, not binary) then combine to yield a complex phenotype. It’s true that in this approach some weights contribute more to the overall facial variance (at least in extant guenon species) than others, and the parent that happens to contribute these weights (if the parents vary substantially along this dimensions) could have a greater impact on the phenotype. But that’s true in genetic inheritance too. This should only pose a problem in our simulations if there were systematic differences in which parent (mother v. father, for instance) contributed each weight, which is not the case. We have clarified these arguments in the text (lines 175-184; lines ESM 175-185). We have also realized that using terms like ‘features’ and ‘feature vector’ to refer to locations in face space, while common in computer science, may lead some readers to mistakenly assume that guenon face space encodes distinct holistic facial features individually (e.g., one ‘feature’ encodes nose spots, another eyebrow patches, etc.), when in reality each dimension of face space is itself a complex axis of variation involving the whole face. To avoid confusion, we have clarified this distinction (ESM lines 85-87) and removed references to facial ‘features’ (we now only refer to face space weights).

It's interesting that the authors decided to model a 15-dimensional “trait”, but this adds a lot of complexity to the model that I think is unexplored and its impact on model outcomes probably overlooked. Adding dimensions can lead to strange evolutionary dynamics in trait space (see for example Doebeli & Ispolatov Am Nat 2016). Also, dimensions are modelled as independent traits, but is it biologically realistic to assume that, for example, selection by assortative mating can operate effectively on 15 traits to lead to a significant divergence between two hybridizing populations? Perhaps this is the reason the authors see populations collapsing unless hybrid fitness is zero? I think that probably this deserves to be further explored.

We thank the reviewer for these interesting comments. We model evolution in a complex multidimensional face space, but ultimately this is encoding a single phenotype (facial appearance). This is a complex phenotype, but each axis of variation is not a separate trait. A more complex analysis of guenon diversification that involves many phenotypic components might involve simultaneously modelling guenon faces, behavioural repertoires, habitat use, etc. to attempt to identify combinations of these variables that promote co-existence, but this is not our goal here. We are also not looking for optimal solutions within face space, unlike some previous work which aimed to model divergence using competition models (e.g. Doebeli & Ispolatov 2017 Am Nat, Doebeli & Ispolatov 2014 Evolution). Furthermore, in order for faces to diverge, selection doesn’t need to act on all 15 axes of variation; diverging on just a few (or even 1, especially with fewer co-evolving populations) would produce unique faces which

would be discriminable across populations and further apart in face space. Precisely how the face space is used and whether there are optimal configurations for coevolving populations would be a very interesting question for future research, but is not the focus here. We have revised the manuscript to clarify these points (ESM lines 64-96).

And this takes me to the next important points. It is surprising that only for 0% fitness the process diversified in sympatry (Table 1). Could the authors provide more detail on this? I wonder how the construction of the model itself, and the choice of parameter values for the simulations could render it insensitive to diversification even when hybrid fitness is just 10% (but see above). Previous models and empirical work showed that diversification in sympatry with gene flow under assortative mating is possible. How are these results explained with respect to this previous evidence? (by the way I think references should be expanded). Also, that diversification in sympatry with assortative mating is lower than with no mate choice is an unexpected result that needs explanation (Table 1). Also this might have to do with parameter choice (as above). The model seems insensitive to different levels of hybrid fitness. Perhaps assortative mating is not strong enough to counteract the high hybrid fitness. But this could be an artifact of the parameter space chosen. I think that further explanation and discussion of these results in this regard is needed.

We thank the reviewer for this insight, which has substantially improved this study. We have added 2% and 5% hybrid fitness to our simulations and updated our methods for characterizing face pattern diversification (see above) to present a more comprehensive and nuanced analysis. We now show that the likelihood of evolving faces that can be reliably clustered by population is inversely proportional to hybrid fitness, but still possible when hybrid fitness is above zero. Our results now show that positive assortative mate choice leads to the most facial diversity (i.e., the highest clustering accuracy across populations); please see above (Referee #1, point #5) for an explanation for our results showing that the greatest distances between populations are associated with no mate choice. We relate our results to previous analyses of diversification in sympatry in the discussion (lines 322-323).

I think that Fig 3 and 4 might reveal that the model could be badly parameterized or constructed. As soon as species enter into sympatry, and mate choice is allowed, populations immediately collapse to what seems to be zero variability. In the case of 0 hybrid fitness, different phenotypes are maintained (with no intrapopulation variability), but this parameter value not unexpectedly forces the model to do so. When fitness > 0 , populations immediately collapse into a single phenotype, revealing that the parameter values are probably too high to reveal a response from the model. It seems to me that a better equilibrium between “drift” by mutation, and selection by mate choice could be looked for through appropriate parameter values. Some discussion is granted, and I'm worried that the author's conclusion that zero hybrid fitness is needed to maintain diversity might be an artifact from model choices.

We have now run additional simulations modelling lower values of hybrid fitness (2% and 5%), which show that some diversification does indeed occur at these levels. We have updated the figures, and the original Figures 3 and 4 have been removed (per the reviewer's suggestion; see below). The new Figures 3 and 4 now show how, in our updated results, hybrid fitness greater than zero can still lead to clustered populations and increased intrapopulation distances.

184 (...) we also assessed whether further diversification (character displacement) occurred after secondary contact in sympatry. We considered this to have occurred when mean distances between populations increased by at least one standard deviation during evolution in sympatry.

How and why was this threshold determined? I think that what happens after secondary contact should be compared to the outcome of the null process (i.e. allopatry), not to some arbitrary value. E.g. are distances at generation x larger for a process with sympatry than with pure allopatry?

We thank the reviewer for this important comment. We have now updated the analysis of population diversification, which is no longer based on an arbitrary threshold. Instead, we use cluster analysis to assign individuals to populations based on their location in face space, and measure population diversification as the proportion of correctly assigned individuals. This updated approach has provided critical insights and significantly improved our study. Our goal here is to understand the persistence of discrete guenon species in sympatry, given that they are characterized by high degrees of sympatry, are commonly found in mixed-species groups, and are often capable of hybridizing; therefore, we think a comparison to pure allopatry is less relevant than comparing across sympatry scenarios.

251 Overall diversification was also observed in three scenarios where hybrid fitness was 10%, all occurring under assortative mate choice and with 50% of evolution occurring in allopatry.

This does not seem to be what Table 1 is showing? Please clarify

We apologize - this analysis has been updated, and this sentence no longer exists in the text.

A relatively less important point is about male quality. Male quality is incorporated into the model but then how it impacts (or not) model outcomes is not mentioned or discussed. How does male quality itself evolve? There is strong selection for male quality so I'm not sure how male quality impacts the outcome of the model. Initial values are drawn from a uniform distribution? What's the average value after, say 10000x generations? I think male quality should be a relative value recalculated at each generation, otherwise the mean (absolute) value tends to evolve towards values of 1, which I don't think makes a lot of sense. In modeling the "devil is in the details" as people like to say, so in some cases adding "biological realism" adds little in terms of insights gained but a lot in complexity and interpretability of the model. I would like to see more detail and discussion on this.

We have included male quality simply as a mechanism for implementing male reproductive skew, and we were not interested here in the evolution of male quality per se. Male quality increases consistently across evolutionary time, as expected, since we do not impose any costs on quality in this scenario. Male quality terms are not bounded by 1, so there is no ceiling effect. In each generation, each males' likelihood of mating is determined based on his relative quality, which is equivalent to rescaling the quality term each generation. We have clarified that the role of male quality is not a key

variable of interest in this study as follows: “Male quality is included only as a mechanism for implementing male reproductive skew and is therefore not subject to costs or constraints in simulations, or to statistical analysis” (ESM lines 116-118; see also lines 156-157).

Also, at the Supplement, line 108, it reads: “ (...) with each phenotypic parameter (face space feature weights and quality or bias terms) having a 0.5% chance of mutating.”
do you mean 0.5 instead of 0.5%? That would be a probability of mutating of 0.005 per generation, which seems to be very low.

This was not a typo. However, we have now increased the mutation rate to 1% (line 191). We also note that this is the probability of mutation for each face space dimension; summing across all 15 dimensions yields a 15% probability of mutation in appearance for each offspring. We have clarified this in the revised version of the manuscript (ESM lines 190-192)

Minor points

Line 78

or lack of genetic variation

It is true that similarity in any given characteristic could be driven by lack of genetic variation. However, we find it unlikely that this would be the cause of the general pattern of stereotyped faces across the guenon clade. This sentence merely suggests stabilizing selection as a cause of stereotyped face patterns within guenon species and does not rule out other potential causes of low within-species variation. Nonetheless, to address the reviewer’s comment, we have updated this sentence to read “Within species, guenon faces are highly stereotyped, with minimal variation associated with age, sex, or seasons [37,39], suggesting stabilizing selection or a lack of genetic variation within species” (lines 75-77).

- A very brief overview of eigenface decomposition would be useful for people not familiar with it.

We have added the following text describing eigenface decomposition: “...a technique that uses principal components analysis to identify key axes of variation in aligned face images (‘eigenfaces’) and has correlates in mammalian visual processing systems” (lines 117-119); “...for instance, the first dimension of guenon face space broadly characterizes overall face colour from dark to light” (lines 121-122).

134 From these initial populations, we simulated 20,000 generations of evolution using a genetic algorithm

What genetic algorithm? Please provide some detail

We have added details explaining the methods implementing cross-generational inheritance in our simulations (lines 175-184; ESM lines 174-187).

146 This models an average face model of face learning and discrimination, in which guenons cognitively encode different species’ face patterns as the mean of all encountered

examples.

Is there any justification for this?

Mating preferences for population average faces could result from a variety of cognitive processes, so we agree that this sentence is a stretch. Our project is not designed to test specific learning models or perceptual processing systems and our results are agnostic to cognitive mechanisms. We have therefore removed this sentence.

174 in this space, distance between points (faces) indicates degree of similarity.

Euclidean distance? Squared Euclidean?

We have kept this sentence since it is just describing face space. Below, we specify that we use Euclidean distance for all metrics (line 209; line 214).

178 We considered populations within a scenario to have diversified when the mean distance between the faces of different populations was at least three standard deviations higher than the mean facial variation within populations.

Why was this cutoff chosen? Observation of behavior of the model? By eye? Please explain.
We thank the reviewer for this important point. We have re-run our analyses of population diversification so that they no longer rely on an arbitrary cutoff. Instead, we use k-means clustering (with k = number of populations) to partition evolved faces into population clusters, and quantify the degree of diversification based on the proportion of correctly clustered individuals. This process is described in the methods (lines 196-207).

207 We compared evolved facial phenotypes across differing conditions using generalized linear mixed models (GLMMs) in a Bayesian framework

As I understand, what was compared was the distance between populations for the different scenarios (i.e. the degree of diversification). Please clarify.

That is correct – we thank the reviewer for catching this! We have updated the text to read “We compared simulation outcomes across differing conditions ...” (line 220).

269 The number of co-evolving populations was also a significant predictor of face distances (posterior mean = 0.0814, pMCMC = 0.026), with distances between faces decreasing with higher numbers of populations.

This might have to do with the fact that all populations start with the same phenotype. Adding populations will tend to reduce the distances as they will be more clumped in trait space.

We agree that results for the number of co-evolving populations can be driven by things like crowding in phenotype space. However, in our new analyses we no longer find this pattern. We have added a brief discussion of how the number of populations may interact with the evolution of diversity to the discussion section, including a description of how phenotypic clustering might influence our results (lines 402-415).

296 (...) further diversification in sympatry and the evolution of stereotyped (within species) and distinctive (between species) faces were only observed under positive assortative mate choice.

This seems to be conflicting with results shown in Table 1 and described previously?

This statement no longer exists in our updated text.

- The main text describing the model and simulations is almost a trimmed copy of the text in the supplement. Extra details provided there are hard to extract as the information is repeated. **We thank the reviewer for this comment. We have edited both the main text and the supplement to move more details to the main text and reduce the overlap between the two (lines 128-185; ESM lines 30-196).**

- It would be interesting to see how interpopulation distances change through time for the different scenarios. A plot showing this should be simple to produce and is likely more informative than a table with arbitrary cutoff values for “diversification” or the figures with the faces or independent plots for the first three features. Faces are nice so I would keep them, but seeing the differences between the different outcomes is difficult, adding some sort of plot of distance through time would greatly help interpretation.

We appreciate this advice. We have added two figures: one depicts the proportion of correctly clustered faces across different mating patterns, degrees of hybrid fitness, and numbers of co-evolving populations (new Figure 3); the second depicts the evolution of interpopulation distances across mating and hybrid viability scenarios, as suggested by the reviewer (new Figure 4).

300 (...) these variables may play a key role in driving phenotypic diversification in one of the most speciose and diverse primate radiations.

Actually it seems that according to the model only zero fitness hybrids play a role in maintaining diversity after secondary contact.

Given that in our revised manuscript we now have updated results showing that positive assortative mate choice is associated with higher clustering accuracy across populations, and that female mating biases are associated with greater distances between populations, we have left this sentence unchanged.

We would like to finish by reiterating our gratitude to the reviewers for their thoughtful and constructive feedback, which has improved the manuscript substantively.

Appendix B

Dear Editors,

Thank you for your consideration of our manuscript, “Simulated evolution of mating signal diversification in a primate radiation” (RSPB- 2022-0734). We are pleased that Proceedings B has elected to publish this manuscript, pending minor revisions.

We have made the requested changes to the manuscript; below are our responses to reviewer comments, with our responses in bold.

We look forward to moving ahead with this publication.

Best wishes,

Sandra Winters & James P. Higham

Associate Editor

Board Member

Comments to Author:

The authors did a careful job addressing the referee's comments and I believe the results is an interesting and well written paper.

Many thanks to the editor for a smooth and constructive review process.

Reviewer(s)' Comments to Author:

Referee: 2

Comments to the Author(s).

I commend the authors for this meticulously revised version of their manuscript. It is a pleasure to review a properly structured revision, marked and with detailed responses. All my concerns have been properly and successfully addressed and now I consider the MS ready for publication.

We appreciate this feedback, and thank the reviewer for their previous comments, which substantially improved this manuscript.

Just caught two little things on the fly

Line 163 "based on a female rota" is this correct?

This is correct, but perhaps confusing. We have clarified this text, which now reads “by cycling through females...and pairing each with randomly drawn males...” (lines 163-165).

Line 170 missing punctuation

We have corrected this typo (line 170).